# Dietary AGEs as Exogenous Boosters of Inflammation

**DOI:** 10.3390/nu13082802

**Published:** 2021-08-16

**Authors:** Ma. Eugenia Garay-Sevilla, Armando Rojas, Manuel Portero-Otin, Jaime Uribarri

**Affiliations:** 1Departamento de Ciencias Medicas, Universidad de Guanajuato, Leon 37320, Mexico; marugaray_2000@yahoo.com; 2Departamento de Ciencias Preclínicas, Facultad de Medicina, Universidad Catolica del Maule, Talca 3480005, Chile; arojasr@ucm.cl; 3Departamento de Medicina Experimental, Facultad de Medicina, Universidad de Lleida, 25196 Lleida, Spain; manuel.portero@mex.udl.cat; 4Renal Division, Department of Medicine, Icahn School of Medicine at Mount Sinai, New York, NY 10029, USA

**Keywords:** low-grade inflammation, inflammatory diet, low AGE diet, Mediterranean diet, DASH diet, dietary inflammatory index (DII), RAGE, matrix glycation

## Abstract

Most chronic modern non-transmissible diseases seem to begin as the result of low-grade inflammation extending over prolonged periods of time. The importance of diet as a source of many pro-inflammatory compounds that could create and sustain such a low-grade inflammatory state cannot be ignored, particularly since we are constantly exposed to them during the day. The focus of this review is on specific components of the diet associated with inflammation, specifically advanced glycation end products (AGEs) that form during thermal processing of food. AGEs are also generated in the body in normal physiology and are widely recognized as increased in diabetes, but many people are unaware of the potential importance of exogenous AGEs ingested in food. We review experimental models, epidemiologic data, and small clinical trials that suggest an important association between dietary intake of these compounds and development of an inflammatory and pro-oxidative state that is conducive to chronic diseases. We compare dietary intake of AGEs with other widely known dietary patterns, such as the Mediterranean and the Dietary Approaches to Stop Hypertension (DASH) diets, as well as the Dietary Inflammation Index (DII). Finally, we delineate in detail the pathophysiological mechanisms induced by dietary AGEs, both direct (i.e., non-receptor-mediated) and indirect (receptor-mediated).

## 1. Introduction

In light of the frequent involvement of inflammatory mechanisms in human diseases, in this paper we offer a short overview of what has generally been considered a relatively minor contributor to inflammation, that is, dietary compounds, and especially the effect of cooking techniques to prepare them. Although dietary compounds may be considered less important than pathology-specific etiological (i.e., causal) factors of inflammation in non-transmissible diseases, it is important to point out that we are exposed to food several times every day over a lifetime. Thus, modification of their contribution, whatever its burden is, should constitute a major driver of health maintenance.

### 1.1. Dietary Modulation of Inflammation

The development of both age- and diet-related non-communicable diseases (NCDS), including metabolic syndrome, cardiometabolic diseases and many cancers, among others, has been associated with low-grade inflammation, characterized by persistently elevated concentrations of circulating pro-inflammatory cytokines [1,2,3]. Diet is an important modulator of chronic and systemic inflammation [4], particularly Western-type dietary patterns, including ultra-processed foods, foods high in fats, refined carbohydrates, and protein, and foods low in fiber, vitamins and minerals. This type of diet can influence normal physiology and can, therefore, affect health by promoting weight gain, with pathological changes in lipids and energy metabolism, evoking a state of chronic metabolic inflammation [5]; whereas diets such as the Mediterranean diet, high in fruits, vegetables, fish and olive oil, are considered a healthier diet, and are associated with lower levels of inflammation [6]. Many NCDs are, to a large extent, preventable. Modifying lifestyle-related risk factors, including an unhealthy diet, may be very important [7,8,9,10].

#### 1.1.1. Dietary Advanced Glycation End Products

AGEs are a heterogeneous group of compounds that were first recognized within biological systems. AGEs are also formed in foods, and their formation is highly dependent on cooking methods, as described below. Only about 10–30% of AGEs present in food are absorbed into the systemic circulation [11]. The mechanisms of the gastrointestinal absorption of AGEs, which have not been fully elucidated, depend on both the hydrolysis of glycated proteins and absorption of the resulting free amino acids or small peptides containing the AGEs [12]. The majority of food AGEs escape digestion and absorption and are delivered to the colon, where there is increasing evidence that they may modify local microbiota metabolism and modulate gut integrity. This local action of unabsorbed food AGEs in the colon may be an important component of the resultant overall pro-inflammatory actions of these compounds in the body [13].

Food-derived AGEs (dAGEs) contribute substantially to the systemic burden of AGEs, and high intake of dAGEs has been linked in both humans and mice to high serum AGEs, oxidative stress, and inflammation, together with reduced innate defenses and insulin resistance [14,15,16,17,18]. They also play an important role in the cause of a number of health disorders [14,19,20,21]. Dietary AGEs induce disease by at least two general mechanisms, one direct (i.e., non-receptor-mediated) and the other indirect (receptor-mediated) (Figure 1).

AGEs produce biological effects through two general mechanisms. On the one hand they crosslink proteins, directly altering their structure and, therefore, their function. For example, glycation of collagen fibers in the skin and vessel walls changes the characteristics of these proteins and their holding tissues as explained within the manuscript. On the other hand, AGEs acting through receptor and non-receptor mechanisms activate different cells leading to increased OS and the release of pro-inflammatory cytokines.

Western diets contain an abundance of dAGEs because of the application of heat in most culinary techniques and the widespread processing of food. Cooking with high heat under dry conditions, such as grilling, leads to significant formation of AGEs, especially in animal-derived foods, which are also rich in fats, while using lower temperatures and high water content, such as stewing, poaching and boiling, decreases the food AGE content. As a result, a piece of meat of the same weight, but exposed to different cooking methods for the same period of time, would have very different AGE content. The essence of introducing a low AGE diet is to make changes in culinary techniques involving temperature, duration, moisture, pH and cooking surfaces that decrease the formation of AGEs during cooking, independent of their macronutrient composition [14]. A few databases with the content of AGEs in a large number of food items have already been published [14,22] which (1) enable the estimation of dietary AGE intake if the method of cooking is described in the food diary, and (2) advise on how to follow a low AGE diet while maintaining approximately the same macronutrient intake.

In a recent narrative review, we presented data from several small interventional trials looking at the effects of dAGE restriction ranging from a few weeks up to 1 year duration. These trials consisted essentially of changing the way of cooking food without modifying caloric and macronutrient intakes [12]. These trials demonstrated that a low AGE diet is associated with a reduction of circulating AGE markers such carboxymethyllysine (CML) and methylglyoxal (MG)-derivatives, as well as markers of inflammation and oxidative stress. Moreover, in patients with diabetes mellitus and metabolic syndrome, the low AGE diet had an additional effect of decreasing HOMA-IR, an index of insulin resistance. A 2017 meta-analysis of 17 published randomized controlled interventions including 560 participants showed that consumption of low AGE diets reduced insulin resistance regardless of participants’ diabetic status, and also diminished fasting insulin levels in patients with type 2 diabetes (T2DM) [23]. Low AGE diets compared to high AGE diets also diminished tumor necrosis factor alpha (TNF-α, vascular cell adhesion molecule-1 (VCAM-1)), 8-isoprostane, leptin, circulating AGE levels, receptors for AGE, and P66^shc^, and increased adiponectin levels and sirtuin-1 in peripheral mononuclear cells in individuals with and without T2DM [23].

A recent study in Iranian patients with metabolic syndrome investigated the effects of 8 weeks of an AGE restricted diet and found a significant decrease in mean serum levels of CML, HOMA-IR, TNF-α and malondialdehyde compared to the regular AGE diet [24]. A randomized crossover study in two four-week interventions in 51 participants without type 2 diabetes tested the hypothesis that a diet high in red and processed meat and refined grains (HMD) would increase inflammatory markers and AGEs when compared with a diet high in dairy, whole grains, nuts and legumes (HWD). The authors demonstrated that the HMD group presented an increase of carboxyethyllysine (CEL), another AGE present in foods, but not CML, when compared to the HWD group [25].

#### 1.1.2. The Mediterranean Diet

The Mediterranean diet is recognized to be a “healthy food” dietary pattern and high adherence to it is associated with a lower incidence of chronic diseases and lower physical impairment in old age [26]. The Mediterranean lifestyle consists mainly of extensive use of a plant-based cuisine using vegetables, fruits, cereals, nuts, and legumes, most of them cooked by adding substantial amounts of olive oil, with moderate use of fish, seafood or dairy, and limited intake of meat and alcohol (mostly red wine) [27]. This diet is low in saturated fats and animal protein and high in antioxidants, fiber and monounsaturated fats, and exhibits an adequate omega-6/omega-3 fatty acid balance. The main compounds that may explain the health benefits of this dietary pattern are antioxidants, fiber, monounsaturated and omega-3 fatty acids, phytosterols and probiotics [28]. Clinical trials have confirmed the favorable influences of this diet on the risk for metabolic syndrome, obesity, type 2 diabetes mellitus, cancer, and neurodegenerative diseases [29]. 

Analyses on the population of the Spanish Prevención con Dieta Mediterranea (PREDIMED) study showed the Mediterranean diet diminished the expression of pro-atherogenic genes [30], cardiovascular risk surrogate markers such as waist-to-hip ratio, lipid fractions, lipoprotein particles, oxidative stress, and markers of inflammation [31,32], as well as reducing the risk of developing metabolic syndrome [33] and type 2 diabetes [34].

The Mediterranean diet could provide a model for the reduction of dAGE, since it provides a low dAGE content, with the consequent reduction of their circulating levels and oxidative stress and inflammation in both elderly adults and metabolic syndrome patients [35,36]. It is important, however, to point out that a low AGE diet allows for the intake of more foods than the Mediterranean diet, as long as they are cooked in the low-AGE way.

#### 1.1.3. The Dietary Approaches to Stop Hypertension (DASH) Diet

The DASH dietary pattern emphasizes the intake of fruit, vegetables, fat-free/low-fat dairy, whole grains, nuts and legumes, and limits total and saturated fat, cholesterol, red and processed meats, sweets, added sugars, and sugar-sweetened beverages. This diet was originally developed to treat hypertension without medications and demonstrated a clinically meaningful blood pressure-lowering effect [37]. In addition to reducing blood pressure, randomized controlled trials have shown the DASH dietary pattern is also associated with reduced low-density lipoprotein cholesterol (LDL-C) among other cardiometabolic risk factors. This diet is also associated with reduction in diabetes and cardiovascular mortality in prospective cohort studies [38,39,40]. Systematic reviews and meta-analyses showed that the DASH diet is associated with lower incidence of cardiovascular disease and better blood pressure control in people with and without diabetes [41]. A systematic review of a total of six controlled trials involving 451 middle-aged participants found that the DASH dietary pattern did not have an effect on C-reactive protein [42]. 

Since the low AGE diet involves only the way of cooking and not a modification of the actual foods consumed, it can easily be integrated into the two other meal patterns described above. A high intake of plant-derived foods is a common factor for the Mediterranean, DASH and plant-based diets, although the actual percent of plant-derived food in each will vary. The low AGE diet, although not a plant-based diet by itself, also emphasizes the use of fruits and vegetables. The low AGE diet may be important in relation to the other dietary patterns when considering the cooking technique used to prepare meats and other animal-derived food sources. A low AGE diet emphasizes three points: (1) decreased intake of high AGE-containing foods (based on existing databases), not by decreasing intake of specific macronutrients, but only through culinary technique modification: temperature, duration, moisture, pH, and cooking surfaces; (2) increased intake of fresh food, which is high in polyphenols and antioxidants, and (3) incorporating the use of herbs, spices, and condiments, which will improve the taste of food and may also have an antiglycation effect (i.e., curcumin, cinnamon, parsley, thyme, and clove). From this perspective, the AGE-less way of cooking might add some extra benefit by further reducing potential oxidants and pro-inflammatory factors in the intake from the other three dietary patterns, although this remains to be proven. 

#### 1.1.4. Dietary Inflammatory Index (DII)

The main objective in creating the original dietary inflammatory index (DII) in 2009 was to develop a tool that would allow the classification of a subject’s diet within a spectrum from maximally anti-inflammatory to maximally pro-inflammatory [43]. The inflammatory capacity of the diet is assessed by first estimating the intake of forty-five micro- and macronutrients, multiplying each by an overall inflammatory effect score, and then comparing them with standard reference data from around the world. The score is globally ranged from a minimum of −8.87 (the most anti-inflammatory diet) to a maximum of +7.98 (the most pro-inflammatory diet) [44]. 

The DII has been studied in various diseases to test the hypothesis that dietary inflammation is a determinant of risk and mortality from NCDs [45]. For example, a meta-analysis showed that a pro-inflammatory diet is associated with increased risk of CVD and CVD mortality [10]. Data from two large prospective studies (n = 18,566 participants) provide evidence that inflammatory mechanisms link an unhealthy diet and premature death. The authors also showed the findings of a meta-analysis with 10 studies and reported a 23% increased risk of all-cause mortality when comparing the lowest vs highest DIIs [7]. The relationship between metabolic syndrome and DII has been controversial with several studies showing a positive association [8,46,47], while in other reports such an association was not observed [48,49,50,51]. 

Trying to integrate the concepts of dAGEs with DII may be difficult, since the latter assigns a score to a series of specific food items independently of the way food is cooked. The opposite is true with the low AGE diet, in which the inflammatory contribution of any food is mainly determined by the way the food is cooked. For example, a person consuming a given amount of chicken will receive a certain fixed DII, while the dAGE score will change dramatically whether the chicken is stewed vs broiled or fried. Whether dietary AGEs are better or worse than DII in scoring an inflammatory diet needs further research.

## 2. Mechanisms of Cell/Tissue Damage Induced by AGEs

### 2.1. Dietary AGEs and Extracellular Matrix Modifications

Exogenous (dietary) AGE-induced extracellular matrix (ECM) modification is one of the potential mechanisms linking dAGE intake and inflammation-related, non-communicable diseases, such as atherosclerosis. Regarding endogenous AGE formation, it is clear that due to their relatively long half-life, ECM proteins are preferential targets of AGE formation [52]. Both in the context of diabetes mellitus-derived hyperglycemia and during aging, some of the AGEs are non-enzymatic crosslinks that contribute to increased rigidity of arterial and connective tissues. Moreover, the modification of shorter-lived proteins, such as apolipoproteins, has also been described [53]. Therefore, the role of endogenous AGEs in the initiation and progression of inflammation-related pathologies (such as atherosclerosis) both via modification of short-lived (e.g., apolipoprotein) and long-lived proteins (ECM) has been demonstrated. Thus, in terms of ECM modification, we propose that, in addition to the endogenous pool of AGEs, dietary AGEs increase the global burden of ECM-AGE, thereby contributing to chronic inflammation. 

ECM-AGE modifications can result in increased macrophage-monocyte chemotaxis [54]. AGE-rich ECM could also decrease the effects of nitric oxide, a central compound in vasodilation [55]. Apart from atherosclerosis, AGE accumulation in ECM is also considered relevant in diabetes mellitus (DM) complications, since it is known to increase the risk of retinopathy and renal failure [56,57]. In diabetic human tissues with high content of ECM, such as intervertebral discs, AGEs also accumulate to a greater degree and are associated with increased matrix-degrading enzymes [58]. In rats, DM has been associated with degenerative ECM changes, including loss of glycosaminoglycan and ECM stiffening, attributed to the accumulation of AGEs [59]. Of note, there is ample evidence that mice with type I DM have increased structural disruption of ECM and related pro-inflammatory cytokines. Thus, AGE links hyperglycemia with ECM structural disruption and the release of pro-inflammatory cytokines [60]. 

The role of ECM in inflammation is far more sophisticated than as a mere cellular scaffold. Thus, it is considered a key participant collaborating in the perpetuation or the resolution of the inflammatory situation. Basement membranes (BM), thin networks of highly crosslinked glycoproteins, and the loose fibril-like interstitial matrix are the two basic forms of ECM. Four major protein components constitute BM: collagen type IV, laminin, nidogen, and proteoglycan perlecan heparan sulfate. Of these, collagen and laminin of type IV can only self-assemble to form networks. Furthermore, specialized ECM structures, which combine the characteristics of both BM and interstitial matrix, form the secondary lymphoid organ reticular fiber network and share characteristics with the provisional matrix that takes shape at injury sites. During wound healing, this provisional matrix can be well populated by immune cells. Specific signals to cells modulate the basic functions essential for the early stages of inflammation, such as migration of immune cells into inflamed tissues and differentiation of immune cells. In chronically inflamed tissues, aberrant ECM expression and ECM fragments derived from tissue-remodeling processes may affect the activation and survival of immune cells, thereby actively contributing to the immune responses of these sites. 

The accumulation of AGEs in tissues is a natural consequence of aging, but it is accelerated in conditions such as DM and renal failure and, importantly, it can be increased by the consumption of unhealthy diets where the formation of endogenous AGEs is enhanced, such as in high fructose diets [61] and/or the consumption of exogenous AGEs produced in high-temperature processed diets [11,62,63]. Regarding dietary AGE sources (i.e., thermally processed foods), about 10 percent of the AGEs in the diet are absorbed and released into the bloodstream via the intestine [11,62]. In several tissues, including the liver and kidney [64], dietary AGEs have been shown to be sequestered directly [64,65]. However, it is rather difficult to obtain an estimation of this amount, as AGE and AGE-related molecules are a family with a vast heterogeneity, especially in human beings. Over the past 20 years, the consumption of Western diets consisting of highly processed foods has been increasing alarmingly, with simultaneous increasing incidence of obesity and DM [66]. Dietary AGEs (and most probably AGE precursors) are a probable source of crosslinking and catabolism in ECM. Thus, it can be proposed that both AGEs and their precursors are highly oxidizing compounds that can accumulate in tissues from endogenous (i.e., hyperglycemia) and exogenous (i.e., high dietary AGE content) sources. Recently, it has been shown that intake of food enriched with a specific AGE precursor (methylglyoxal) accelerates age-related vertebral bone loss in aging pre-diabetic mice [65], affecting ECM. When further examined, ingestion of dietary AGEs induced sex-dependent, inferior biomechanical bone quality in young female mice (6 months) [67,68]. The first evidence that dietary AGEs induced ECM changes, in which the vertebral structure and function were directly affected, was provided by these studies [65]. These studies demonstrated that consumption of chow treated with high temperature (to increase exogenous AGEs), especially in female mice, resulted in accumulation of AGEs in ECM, as shown by direct analyses of intervertebral discs. This diet was associated with functional changes of this organ as a result of altered ECM, including increased compressive stiffness and increased torque forces range. Indeed, significant changes were observed in the quality of collagen, and the significantly increased crosslinking of AGEs was most likely responsible for the biomechanical changes observed in these young animals. Augmented modification to collagen was also observed and will probably increase the risk of failure as the affected organ increases in age. 

As mentioned above, in preclinical models the consumption of diets with high AGE content (i.e., thermally processed or enriched with fructose) leads to the accumulation of AGEs within tissues with a high ECM content, such as intervertebral discs. Compared to low AGE diets, the AGE content of ECM located in intervertebral discs in mice fed with high AGE content diets increased by about 150 percent [68]. From a structural point of view, one of the mechanisms of AGEs in altering tissue structure and function is the action of crosslink proteins. For instance, pentosidine is a crosslink between lysine and arginine residues belonging to the same or to different protein molecules [69]. In tissues such as articular cartilage, stiffness and brittleness are associated to crosslinks of long-living proteins, such as aggrecan and collagen [70], both present in bone and other ECM rich tissues [71]. In these organs, AGE collagen crosslinks accumulate, implying a loss of ECM integrity and advancing the pathogenesis of the degenerative process [72]. In addition, in vitro ECM glycation specifically increases tissue rigidity and brittleness [73]. Collagen damage can be caused by increased crosslinking due to glycation [74,75,76] resulting in structural disorder of the collagen fibrillar array. Thus, glycation has been shown to disrupt the organization of collagen in several ways: through increased fibril packing and diameter and by altering fibril organization [77]; AGE adducts can twist and distort the collagen fibrillar array [76]; and AGEs can alter molecular packing by increasing the distances between collagen chains and molecules [78]. Indeed, in preclinical models the data indicate that biomechanical changes are partly driven by AGE crosslinking in the extracellular matrix, although it cannot be discounted that diet-derived AGES or AGE precursors could impinge oxidative modifications that result in damage to collagen, which is likely to increase the risk of mechanical injury in older mice or under higher load conditions [68]. It has been suggested that crosslinking with AGE can lead to mechanically induced collagen triple-helical structure damage derived from accelerated enzyme degradation, transmitted through the intermolecular AGE crosslinks [79]. The contribution of AGEs also goes further than the biomechanical changes induced in ECM, comprising changes in cellular responsese toi AGE-modified ECM, in particular the generation of a fibrosis-prone gene expression profile, which is widely documented in many complications of diabetes [80,81]. Thus, AGEs induce enzymatic activity and catabolism of the matrix, mediated by binding to the AGE receptor (RAGE) [82]. Previously data have suggested an important role for the AGE/RAGE pathway [55] in DM and AGE-induced ECM degeneration, but there is still scope to further investigate the role of this interaction, particularly in relation to dietary AGEs in humans (see next section in this chapter). Moreover, it has been described that dAGEs do not always bind to RAGE [83] so other RAGE-independent mechanisms can also operate. Some dAGEs, and particularly CML, are stable, chemically unreactive compounds that cannot damage the tissue by direct interaction with tissue proteins. Apart from CML, other Maillard reaction-generated products, such as dicarbonyl compounds (e.g., glyoxal, methylglyoxal and 3-deoxyglucosone), are also present in diets with high AGE contents. These reactive compounds could perform a direct modification of ECM [84]. Nonetheless, there is controversy on whether these reactive compounds could contribute to AGE burden, since other researchers have shown that dietary dicarbonyls are unlikely to directly contribute to the endogenous pool of these compounds [85]. It has also been suggested that high-fat diets induce in vivo dicarbonyl formation [85,86]. Therefore, the increased level of endogenous AGEs observed in in vivo studies offering a roasted, high-fat diet might be partially explained by the high-fat induced formation of dicarbonyl compounds. 

Further indications regarding dAGE health effects could be provided by studies in tissue distribution and renal clearance of labeled dAGEs. Rats supplied with AGEs derived from bread crust [87,88] exhibit increased content of CML in tissues rich in ECM, such as tail tendon and heart. Similarly, mice supplied with CML-enriched diets also exhibit enrichment of aorta CML concentration [89]. In these later experiments, in comparison with the control diet, the CML-enriched diet led to reduced endothelium-dependent relaxation, in association with increased aortic expression of RAGE and VCAM-1. This was strongly dependent on RAGE, since RAGE−/− mice were protected against these changes [89]. 

Globally, dietary AGE content may be a crucial determinant of the accumulation of AGE crosslinks in tendons and for tissue compliance, relevant to the biological properties of ECM. Tessier et al. [63] unequivocally identified a relationship between dietary CML and its deposition in organs using radio-labeled CML. However, there is no agreement to what degree the accumulation of dAGEs (and which AGE) in tissues could have pathological implications. In this sense, there is a high heterogeneity in serum AGE levels in healthy subjects, indicating that dAGEs might not necessarily be dangerous unless there is a concomitant disease. In this sense, patients with renal disease exhibiting an inefficient AGE excretion mechanism [90,91] have a clear increase in circulating AGE levels. This rise is due in part to the impaired excretion of endogenous AGEs [89] since a low dAGE diet intervention showed a marked decrease in circulating AGEs in these patients [92,93]. Of note, skin autofluorescence, a purportedly indirect assessment of skin AGE, has been recently described as an independent risk factor for cardiovascular complications in individuals with uremia [94]. 

It is noteworthy that very few papers discuss the potential influence of dAGEs in intracellular AGE modification in humans. Despite ample evidence in rodents [95] even for neural tissue modification and validation studies in humans [96], no exhaustive studies have been undertaken for the evaluation of dAGEs as a direct AGE modifier of intracellular proteins. These intracellular modifications [97,98] play a key role in the processes where AGEs (endogenous) have been invoked. They may also play a role in the inflammatory status [99]. However, the fact that dAGEs could enhance cellular oxidative stress and derived signaling, increasing in situ formation of AGE products or highly reactive AGE-precursors such as methylglyoxal [100], gives a further dimension to the already intricate relationship of dAGEs with inflammation.

A very recent paper adds a novel layer of complexity in the dAGE relationship with inflammation. Snelson et al. have shown that long-term consumption of dAGEs leads to changes in intestinal barrier permeability in rodents [101]. This alteration increases innate immune complement activation and local inflammation (especially evident at the renal level), via generation of the pro-inflammatory effector molecule complement 5a (C5a). These data reinforce previous evidence showing increased concentration of several cytokines, such as IL-16, IL-1α, ICAM, TIMP-1 and C5a, in mice fed with high AGE dietary contents [102]. Mechanistically, in vitro data show that AGE-modified albumin cleaves C5 to generate C5a [103]. In humans, lutein treatment induces decreased C5a concentrations in patients with early macular degeneration [104]. Indeed, C5a tissue levels are enhanced in the vitreous of patients with proliferative diabetic retinopathy [105]. However, whether changes in dAGEs impinge C5a variations in humans is still unknown. Of note, a decreased burden of dAGEs changes gut microbiome [106], which is a primary driver in the dAGE-derived innate immune complement activation. Thus, it may be proposed that dAGEs acting either directly on ECM proteins, or indirectly altering intestinal permeability barriers, could change inflammatory status in humans.

### 2.2. Dietary AGEs and Receptor-Mediated Mechanisms

As mentioned previously, AGEs tend to accumulate in a wide range of chronic conditions, such as DM, vascular disease, arthritis, aging, cancer, and neurodegeneration [107,108,109,110]. Apart from endogenous production, dietary AGEs are important contributors to the body’s AGE pool.

Many AGEs-binding proteins have been described so far, most of them involved in the endocytic uptake, breakdown, and removal of AGEs from the circulation [111]. The exception is the protein known as RAGE, which is the central transduction receptor for AGEs. RAGE belongs to the immunoglobulin superfamily of molecules. Its structure comprises a multi-ligand binding extracellular domain, a membrane-spanning domain, and an intracellular carboxyl-terminal domain [112]. The extracellular domain is composed of three smaller domains, one V-type domain with homology to immunoglobulin variable domains, and two C-type domains with homology to the immunoglobulin constant domains [113]. While RAGE is the product of a single gene, alternative splicing generates soluble RAGE, lacking both the membrane-spanning and intracellular domains [114,115]. Additionally, extracellular metalloproteinases can cleave cell surface RAGE leading to an additional soluble receptor form. Independently of the mechanisms involved, soluble RAGE molecules serve as decoys for circulating AGEs and other ligands [116].

The recognition of AGEs by RAGE initiates complex signaling pathways, including reactive oxygen species (ROS), p21ras, erk1/2 (p44/p42) MAP kinases, p38 and SAPK/JNK MAP kinases, rhoGTPases, phosphoinositol-3 kinase and the JAK/STAT pathways, rendering crucial down-stream inflammatory consequences, such as the activation of crucial transcription factors such as NF-κB, AP-1, and Stat-3. The activation of NFkB up-regulates RAGE expression itself, thus generating a positive feed-forward loop rendering more inflammation [117].

At present, the binding capacity of RAGE has been extended to a diverse series of other ligands beyond AGEs, many of which can be defined as pathogen-associated molecular patterns (PAMPs) or damage-associated molecular patterns (DAMPs), leading to the recognition that RAGE is another pattern recognition receptor (PRR). This ligand binding diversity adds more complexity to the intracellular signaling network of RAGE, considering that this receptor shares ligands and pathways with toll-like receptors, particularly with TLR-2 and -4 [118].

The short cytoplasmic tail lacks intrinsic kinase activity, however, compelling data derived from both in vitro and in vivo experiments indicate that the cytoplasmic domain is essential for intracellular signaling, by recruiting adaptor proteins, such as Diaphanous-1 and toll-interleukin 1 receptor domain-containing adaptor protein (TIRAP), or the extracellular signal-regulated kinase Erk1 [119,120,121].

It is noteworthy that the complexity of signaling has been further evidenced by several reports showing RAGE interactions with other plasma membrane proteins, as occurs with the alternative adaptor molecule DAP10, which seems to be crucial for enhancing the RAGE-mediated signaling, as well as the interaction of RAGE with chemotactic G-protein-coupled receptors (GPCRs) on the plasma membrane [122,123]. Furthermore, a cognate ligand-independent mechanism for RAGE transactivation has been recently reported to occur following activation of the AT1R in different cell types [124].

Additionally, RAGE oligomerization has been highlighted as a crucial event in signaling [125] because disruption of the self-association results in reduced mitogen-activated protein kinase phosphorylation and nuclear factor NF-κB activation [126].

At present, a huge body of compelling evidence demonstrates RAGE and its ligands are a major inflammatory driving mechanism in the pathogenesis of many clinical entities where inflammation is an important element in the onset and/or progression of the disease [124,125,126,127,128,129].

Most of the AGE pathological actions are due to the interaction with the full-RAGE anchorage at the cytoplasmic membrane. However, one interesting point is raised by some reports showing the presence of intracellular AGEs, particularly CML and MGO [18,130]. These findings are particularly important because RAGE has also been reported to be located intracellularly and co-localized with endosomes, particularly in T cells from patients with T1DM [131]. Thus, it is tempting to speculate that intracellular AGEs may represent a “danger signal”, which may be then recognized by endosomal RAGE and thus activating intracellular signaling pathways, as occurs with other pattern recognition receptors [132].

In recent years, NLRP3 (NOD-, LRR- and pyrin domain-containing protein 3) has emerged as a crucial intracellular sensor that detects a broad range of microbial motifs, endogenous danger signals, and environmental irritants, resulting in the formation and activation of the NLRP3 inflammasome [133]. Assembly of the NLRP3 inflammasome leads to caspase 1-dependent release of the pro-inflammatory cytokines IL-1β and IL-18, as well as to gasdermin D-mediated pyroptotic cell death [134]. The role of AGES in the activation of NLRP3 inflammasome remains controversial. While some reports support the role of AGEs in the activation of the NLRP3 inflammasome in endothelial cells [135], placental tissue [136], chondrocytes [137], and podocytes [138], other authors have demonstrated that AGEs could impair NLRP3 inflammasome activation [139,140]. Thus, AGEs regulation of NLRP3 inflammasome activity appears to be highly dependent on cell type, and this issue, therefore, requires further investigation. Furthermore, the primary inflammasome research interests have been focused on NLRP3, and the effects of AGEs on the subset of NLRs other than NLRP3 is still a matter of research.

## 3. Conclusions

Diet, an important component of modern lifestyle, plays an important role in increasing low-grade inflammation and oxidative stress, which are thought to be the main mediators of many modern chronic non-transmissible diseases. Although there are many dietary factors that may be associated with inflammation and oxidative stress, we have emphasized here the potential role of dietary AGEs. Despite the wide recognition of AGEs as important factors in the pathogenesis of diabetic complications, the role of AGEs of dietary origin as a factor in human disease has been largely unappreciated. We have presented in this review further evidence of this role and described the many metabolic pathways leading to these effects.

## Figures and Tables

**Figure 1 nutrients-13-02802-f001:**
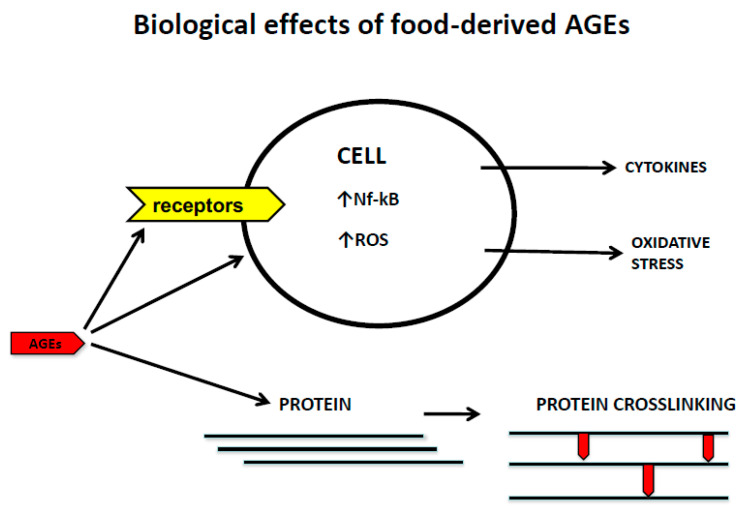
Biological effects of food-derived AGEs. ↑ refers to increased formation.

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
