# Peer review of "Dietary AGEs as Exogenous Boosters of Inflammation"

_nutrients, 2021, doi:10.3390/nu13082802_

Round 1

Reviewer 1 Report

Garay-Sevilla and colleagues review recent literature on the contribution of dietary AGEs to low-chronic inflammatory response underlying the pathogenesis of many human diseases. 

The review is well organized and provide an interesting point of view on the potential beneficial effects of integration of a low AGEs diet to well-established healthy dietary regimens such as the Mediterranean or the DASH. 

I would suggest only minor modifications:

  • when mentioning the existence of databases reporting the AGEs content of most popular food items, it would be interesting to also read about the main specific important findings reported in them, maybe thrugh a table. 
  • to emphasize the possible synergistic effect of an integration of a low-AGEs diet to the Mediterranean, DASH, and Plant-based diets, I would suggest to combine paragraphs 1.3 and 1.4 describing the mediterranean and DASH diet, providing, whether available, information or comments on the AGE content of these dietary regimen, and then underlying the potential of an integration with a low-AGEs approach described in the last part of pargraph 1.4.
  • a relatively novel and intriguing aspect of systemic inflammation induced by dietary behaviours, including the consumption of high dietary AGEs, is the modulation of gut integrity and gut micorbiota, which is reported to widely contribute to the establishment of a systemic inflammation through the activation of TLR4 and NLRP3 also in distal organs. I would suggest to further expand this issue that is now briefly described at the end of paragraph 2.1 by adding more recent literature data on dAGEs and gut modifications and related systemic inflammation.
  • Finally, it would be useful for readers to visualize some of the main described mechanisms for dAGEs-induced inflammation, also involving ECM glycation, through a schematic illustration.

Some typos:

In abstract: [...] but many people are unaware of THE (not THEIR) potential importance [...]

In keywords: matrix glycation is reported as matric glycation

Author Response

Review #1

Garay-Sevilla and colleagues review recent literature on the contribution of dietary AGEs to low-chronic inflammatory response underlying the pathogenesis of many human diseases. 

The review is well organized and provide an interesting point of view on the potential beneficial effects of integration of a low AGEs diet to well-established healthy dietary regimens such as the Mediterranean or the DASH. 

I would suggest only minor modifications:

  • when mentioning the existence of databases reporting the AGEs content of most popular food items, it would be interesting to also read about the main specific important findings reported in them, maybe through a table. 

We appreciate the value of this reviewer’s suggestion, but the creation of such a table goes beyond the scope of this review and we will not follow it. There are several published databases with the content of AGEs in different foods, but we only cited two in this particular review.

  • to emphasize the possible synergistic effect of an integration of a low-AGEs diet to the Mediterranean, DASH, and Plant-based diets, I would suggest to combine paragraphs 1.3 and 1.4 describing the mediterranean and DASH diet, providing, whether available, information or comments on the AGE content of these dietary regimen, and then underlying the potential of an integration with a low-AGEs approach described in the last part of paragraph 1.4.

We appreciate the reviewer’s comments and we have made several changes in the updated manuscript reflecting reviewer’s suggestions.

  • a relatively novel and intriguing aspect of systemic inflammation induced by dietary behaviours, including the consumption of high dietary AGEs, is the modulation of gut integrity and gut micorbiota, which is reported to widely contribute to the establishment of a systemic inflammation through the activation of TLR4 and NLRP3 also in distal organs. I would suggest to further expand this issue that is now briefly described at the end of paragraph 2.1 by adding more recent literature data on dAGEs and gut modifications and related systemic inflammation.

We have followed reviewer’s suggestions and added a few sentences on the subject of unabsorbed dietary AGEs and their potential actions in the colon as well as added some more recent research in the area.

  • Finally, it would be useful for readers to visualize some of the main described mechanisms for dAGEs-induced inflammation, also involving ECM glycation, through a schematic illustration.

We have followed the reviewer’s suggestion and created a Figure 1 reflecting the different actions of dAGEs in the body.

Some typos:

In abstract: [...] but many people are unaware of THE (not THEIR) potential importance [...]

            In keywords: matrix glycation is reported as matric glycation

We have corrected the above typos pointed out by the reviewer

Reviewer 2 Report

In this review the authors have been well summarized the involvement of dietary advance glycation end products in the pathogenesis of many disease.

Minor Revision:

Abstract: Please, write: both direct and indirec way;

Keywords: Please, delete DII and write: Dietary inflammatory index;

Inflammatory mechanisms elicited by dietary AGEs: In my opinion this title is inappropriate because the authors do not mention AGEs;

1.2 Dietary advanced glycation end products.

Food-derived AGEs: The authors should describe what AGEs are. Therefore the subheading “Dietary modulation of inflammation” should be deleted and the information contained therein could be included in the next subheading.

Abbreviations should be defined in parentheses the first time they appear in the abstract and in the main text and used consistently thereafter.

Author Response

Reviewer #2

In this review the authors have been well summarized the involvement of dietary advance glycation end products in the pathogenesis of many disease.

Minor Revision:

Abstract: Please, write: both direct and indirect way;

Done

Keywords: Please, delete DII and write: Dietary inflammatory index;

Done

Inflammatory mechanisms elicited by dietary AGEs: In my opinion this title is inappropriate because the authors do not mention AGEs;

We agree with the reviewer and changed the title

1.2 Dietary advanced glycation end products. 

Food-derived AGEs: The authors should describe what AGEs are. Therefore the subheading “Dietary modulation of inflammation” should be deleted and the information contained therein could be included in the next subheading.

We have followed the suggestion of the reviewers and describe AGEs and change the subheadings

Abbreviations should be defined in parentheses the first time they appear in the abstract and in the main text and used consistently thereafter.

Done

Reviewer 3 Report

This review article is well written and covers important aspects of AGE mediated effects. It will be informative, however, how AGE products produced during cooking is absorbed, and modified in the liver and circulated through the body. 

Author Response

Reviewer #3

This review article is well written and covers important aspects of AGE mediated effects. It will be informative, however, how AGE products produced during cooking is absorbed, and modified in the liver and circulated through the body. 

We have now added a small sentence with this information in the section of description of AGEs in the updated manuscript.